# Bibliometric Analysis: Six Decades of Scientific Production from a Nationwide Institution: Instituto de Seguridad y Servicios Sociales de los Trabajadores del Estado (ISSSTE) from Mexico

**DOI:** 10.3390/healthcare11121725

**Published:** 2023-06-12

**Authors:** Gerónimo Pacheco Aispuro, Ileana Belén Rojas Jácome, Carlos Alejandro Martínez Zamora, Cuauhtémoc Gil-Ortiz Mejía, Christopher Mader, Carlos Castillo Rangel, Alejandro Monroy Sosa, Mario Flores-Vázquez, Octavio Jesús Arroyo Zavala, Rodrigo Ramos-Zúñiga, Guillermo González Garibay, Gerson Ángel Alavez, Ángel Lee

**Affiliations:** 1Department of Neurology, Hospital Ángeles del Pedregal, Mexico City 10700, Mexicoileanarojas97@gmail.com (I.B.R.J.); 2Mexican Faculty of Medicine, Universidad La Salle, Mexico City 06140, Mexico; 3School of Medicine, Saint Luke Escuela de Medicina, Mexico City 11000, Mexico; 4Department of Neurosurgery, Centro Médico Nacional 20 de Noviembre ISSSTE, Mexico City 03104, Mexico; 5Department of Neurosurgery, ISSSTE Hospital Regional Lic. Adolfo López Mateos, Mexico City 01030, Mexico; 6Department of Neurosurgery, ISSSTE Hospital Regional 1° de Octubre, Mexico City 07760, Mexico; 7Department of Neurosurgery, Hospital ISSSTE Tláhuac, Mexico City 13273, Mexico; 8Department of Neurosurgery, Hospital Regional Dr. Valentín Gómez Farías-ISSSTE, Zapopan 45100, Mexico; 9Department of Neurosurgery, ISSSTE Hospital Regional Gral. Ignacio Zaragoza, Mexico City 09100, Mexico; 10Center of Health Sciences, University of Guadalajara, Guadalajara 44100, Mexico; 11Faculty of Health Sciences, Universidad Anáhuac México Norte, Naucalpan de Juárez 52786, Mexico; 12Hospital Ángeles del Pedregal, Mexico City 10700, Mexico

**Keywords:** bibliometric analysis, Scopus affiliation errors, ISSSTE, Mexican scientific research, L0 index, manuscript writing, honorary authorship

## Abstract

Background: This study employed bibliometric analysis to ascertain the research focus areas among a group of Mexican physicians affiliated with the Instituto de Seguridad y Servicios Sociales de los Trabajadores del Estado (ISSSTE). ISSSTE, a healthcare institution catering to a diverse range of diseases, offers a distinctive perspective on the investigated specialties within the realm of health. The primary objective was to identify knowledge gaps in medical care disciplines through a comprehensive examination of scholarly publications. Methods: We retrieved Scopus papers affiliated with “ISSSTE” and saved them as .CSV files. Subsequently, we employed VOSviewer, biblioshiny, and bibliometrix for bibliometric analysis. This enabled us to identify prominent institutions, prolific authors, highly cited researchers, and their respective affiliations. Results: Our analysis identified 2063 publications; the specialty internal medicine accounted for the greatest proportion with 831 publications. Original papers accounted for 82% of the total, with 52% of them being written in Spanish. The majority of scientific output, 92%, originated from Mexico City. The annual production has steadily increased since 2010, peaking in 2021 with over 200 publications. However, papers on prevalent conditions, such as metabolic syndrome, received limited citations, and the L0 index (percentage of uncited items) for all papers is close to 60%. Scopus mislabeled one affiliation, and some cases show a low paper-to-author ratio of 0.5 Discussion: Additional concerns, such as honorary authorship due to excessive authors per paper, and the underlying causes of low citation rates in Mexican publications, warrant further examination. Moreover, our research emphasizes the urgency of bolstering research and development funding, which was consistently below 0.5% of GDP for the past four decades, falling short of legal mandates and international benchmarks. We endorse the establishment of robust research collectives in Latin America to address these challenges, foster regional scientific output, and transition from knowledge consumers to knowledge producers, thereby reducing dependence on foreign technology.

## 1. Introduction

Healthcare services in Mexico are split into public and private sectors. Government hospitals are further divided into three macro-institutions according to the patient’s employer. For a private employee, the Instituto Mexicano del Seguro Social (IMSS) will provide healthcare, but if it is a government official, the Instituto de Seguridad y Servicios Sociales de los Trabajadores del Estado (ISSSTE) will fulfill that role. The Secretaría de Salud will care for self-employed and unemployed patients. Those who can afford a private insurance plan or have out-of-pocket payment capacity will visit private hospitals or clinics [1,2]. Other minor stakeholders are beyond the scope of this manuscript, and The Best Health Care in the World 2022 ranks countries around the planet according to a wide range of factors: care process, accessibility, administrative efficiency, equity, health outcomes [3], infrastructure, health care professionals, and gross domestic product (GDP). Mexico ranked 68th. The ISSSTE covers about 10% of the Mexican population (around 13.7 million) [4,5]. Its mission is to “contribute to improving the levels of wellness of the workers of the State and all their beneficiaries, by providing efficient, effective, and good quality services [4]. In medicine, scientific research is the cornerstone for improving patient care, finding innovative solutions, and making healthcare more affordable to our economic conditions. Innovation is not restricted to cutting-edge technologies for the most complex pathologies, and novel ways of medical care can be found with modest means. According to Article 25 of the General Education Act (“Ley General de Educación”), Mexico should devote 1% of its GDP to scientific research and technological development, and the main stakeholders are public higher education institutions [6,7,8]. According to the World Bank, Mexico’s expenditure in research and development (R&D) has never surpassed 0.5% as shown in Figure 1, and barely approaches 0.3% in 2020 [9,10].

The world’s bibliometric analysis (BA) primarily focuses on academic productivity using published scientific literature, including research articles, books, and conference proceedings, to measure research activities in a specific area [1]. Scientists use BA for various purposes, such as discovering emerging trends in a journal, analyzing article performance, identifying patterns of collaboration, and investigating research constituents such as author details, co-authorship, and citations obtained [2]. BA is also useful for examining the intellectual structure of a particular area in the existing literature. The popularity of BA has grown exponentially with an increasing number of articles discussing its applications [2]. This popularity stems from its ability to highlight both hotspots and neglected areas in research, making it the “Google Maps of medical research,” as previously published [3]. As mentioned in a recent publication [4], bibliometric analysis has been widely overlooked by Mexican authors until recently. In fact, a PubMed search for articles with titles containing the phrases “Mexican publications” or “Publications Mexico” yielded only five results, some dating back as far as 70 years ago. Although medicine is among the most productive disciplines in Mexico [5], a more detailed thematic analysis is still lacking.

Bibliometric indicators offer a way to measure productivity, which refers to the number of publications, and impact, which refers to the frequency of citations at different levels: micro (individuals), meso (hospitals, clinics), and macro (countries, specialties, universities, or large health institutions). These parameters obtained through BA provide a quantitative understanding of the scientific performance of a group, although this concept is controversial.

Mexico has a high degree of decoupling between public and private health providers, and public health services are financed, managed, and delivered by different institutions. Patients with health insurance receive care from various public providers, based on the nature of their employers. The Instituto de Seguridad y Servicios Sociales de los Trabajadores del Estado (ISSSTE) provides care for public servants and the Instituto Mexicano del Seguro Social (IMSS) provides care for other insured individuals. Patients without insurance were primarily treated by the Secretaría de Salud (Ministry of Health) and its hospital network [5]. The structure of the ISSSTE offers favorable conditions for studying its scientific production. Medical needs in the institution are addressed at the primary, secondary, and tertiary levels of care, covering the entire country and encompassing all medical specialties from family medicine to robotic surgery. Furthermore, civil servants in Mexico typically maintain their employment, whereas employees in the private sector may experience rotation due to factors such as recruitment, dismissal, or even company bankruptcies during recessions. This often results in patients with IMSS being lost to follow-up, and long-term research protocols are becoming more difficult to carry out.

Despite the linguistic, cultural, and religious homogeneity shared by the diverse sovereign nations of Latin America, several negative factors can hinder scientific production in the region, such as political instability, corruption, and dependence on commodity exports. These factors make it necessary to conduct bibliometric analyses specific to Mexico, as they can help identify and address issues that are unique to the country and not necessarily representative of the larger Latin American region.

When discussing the low performance of the largest Latin American countries (Brazil, Argentina, Mexico) in scientific research, it is often assumed that their economies are “poor”. It is commonly postulated that their economies exhibit a notable degree of underperformance. The intended meaning is a prevalent belief that underperformance in bibliometric analysis can be mistakenly associated with underperformance in scientific research. Similarly, blaming previous or present administrations is another common reason for the lack of progress in this area. However, we want to show that both assumptions do not have a causal relationship with the poor performance of science and do not fully explain the situation.

The world’s largest economies are organized into two distinct international groups based on the following criteria:

G20 (group 20), and Organization for Economic Cooperation and Development (OECD). Among these, three Latin American countries are members of the G20: Argentina, Brazil, and Mexico, whereas Chile, Colombia, Mexico, and Costa Rica belong to the OECD. According to International Monetary Fund data, ranked by size [6], the Mexican economy holds the 15th position in the world, outpacing wealthy nations, such as Switzerland and the Netherlands. Mexico is uniquely positioned as the only Latin American country and is a member of both the G20 and OECD. However, its investment in research and development (R&D) is alarmingly low compared with its counterparts in both groups (Figure 1). Argentina spends 100% more and Brazil 200% more on R&D, while the OECD’s average spending is a staggering 800% larger than Mexico’s investment [7].

In contrast it is important to note that high-ranking officials in Mexico are aware of the situation. For example, the current Mexican Permanent Representative to the United Nations, Juan Ramón de la Fuente, previously served as Secretary of Health, from 1994 to 1999 and as a rector of the National Autonomous University of Mexico (UNAM, the largest and the second oldest institution in the Americas) from 1999 to 2007. In an interview published in *Nature* in the year 2010, he stated: “Mexico does not have a public policy on research and development. If you review what has happened with science over the past 30 years, it is evident that each government has committed to science to varying degrees, making it difficult to maintain continuity, which is crucial in science. The proportion of the gross domestic product that Mexico dedicates to science and development has not significantly increased over the past two or three decades” [8]. It is worth noting that 13 years later, the situation has hardly improved, as demonstrated in Figure 1, which means that nothing has changed in at least the last 40 years. According to Article 25 of the General Education Act (“Ley General de Educación”) [9], Mexico is legally required to allocate 1% of its GDP to scientific research and technological development. However, according to our World Bank table, this percentage only reached 0.3% in 2020, which theoretically places the country in violation of its laws. Statements by high-ranking officials indicate that this problem has persisted for several decades. It is imperative that tangible and effective efforts are made to propel Mexico to the forefront of the scientific field. We must emphasize that this issue goes beyond a mere “economic disadvantage issue”. Despite having the world number 15 economy, as shown by the Scimago Country Ranking [10], Mexico’s scientific production ranks 26th.

In the last few decades, techniques that measure and analyze the scientific production of any given country and/or institution have emerged as powerful and popular tools [11,12]. Literature production has been growing, so bibliometrics can explore large volumes of data, analyze them on macroscopic and microscopic levels [13], and create a quantitative report with a descriptive purpose: to reflect the state of the research at a specific moment in time and space. This analysis highlights research hotspots, detects research trends, enables comparisons with similar nations, and sets standards for later studies. In the case of Mexico, we may even compare our future performance with the standards defined in the current study.

Bibliometric reports for institutions are now becoming standard in best practices across responsible metric scenarios [11]. Science indicators should encourage the creation of new strategies to improve the scientific production of a country, specifically of prestigious institutions, such as those we are targeting. At the core of bibliometrics, the application of mathematical and statistical methods as the quantitative analysis of the bibliographic features of a body of literature [13] lies in its ability to show what has been investigated and to fill in the gaps, just like any map highlights densely and sparsely populated areas. These data are a starting point for improving our understanding of the psychological, behavioral, and societal aspects that lead to the generation of new knowledge, which is the only purpose of medical research.

National medicine has seldom been analyzed, and previous papers on Mexican performance are narrow, either focusing on a specialty, a hospital, a medical society, or recording very short periods of time. No one has performed a truly “three-dimensional” bibliometric analysis across space (the whole country), time (several decades), and thematic covering (all medical specialties). This type of analysis will set the foundations for further projects that would disclose research hotspots, frontiers, and trends in the field of Mexican medical science.

Bibliometrics also identifies collaboration networks between institutions or even countries and can therefore be a strategy to promote and develop research [14], globalize knowledge production, and use it to improve healthcare quality. As a corollary, the use of scientometric tools identifies publication trends for comparison with commonly encountered problems in patients. These strategies will identify research gaps in primary care and might help promote healthcare improvement focused on the most prevalent diseases treated by ISSSTE hospitals [15]. This highly valuable tool can build a solid foundation for advancing any field and can certainly guide research but has largely been underused in Mexico [16,17]. Ours is a pioneer project analyzing a Mexican institution with nationwide distribution, from inception to the present time, including the present COVID-19 pandemic, and is a foundation for later analyses in our country or in our region.

The purpose of our study was to identify the main stakeholders in Mexican medical research and the frequency with which specialties that address the most prevalent pathologies contribute to the global corpus of medical knowledge. By focusing on the most cited papers in our bibliometric analysis, we will be able to identify the topics that have drawn the attention of the medical community and compare them with the most prevalent health conditions treated by the ISSSTE. It is worth noting that any research project that does not result in publication (i.e., accessible to the public) remains private and is known only to the researchers. Our study aims to fill this knowledge gap by analyzing the scientific output of a large Mexican medical care institution in terms of articles, citations, co-authorship networks, medical specialties involved, and diseases being researched using appropriate keywords. This is the first Mexican, “three-dimensional” BA, encompassing space (the entire country), time (several decades), and thematic coverage (all medical specialties). This will establish a Mexican national standard for productivity in medical research, allowing other stakeholders, such as the IMSS, Secretaría de Salud, and private hospitals, to compare their own bibliometric indicators. Furthermore, our study lays the foundation for future research projects in this area.

## 2. Materials and Methods

### Data Source and Data Extraction

Our study was conducted utilizing Scopus, an extensively employed database in our prior bibliometric investigations pertaining to Latin American research [18], the COVID-19 pandemic [19], epilepsy [11,12], dementia [13], brain tumors [14], neurosurgery [4], and neurocysticercosis [15]. This repository was chosen because it is a comprehensive, diverse, interdisciplinary, and neutral source that provides abstracts and citations with a significant overview of the world’s research output in the field of medicine and other related fields. We included articles authored by at least one researcher with an institutional affiliation to the ISSSTE and published from 1969 to 2021. Utilizing Scopus’ built-in functions, we extracted the following bibliometric variables from each retrieved paper: author, author ID, author affiliation, article title, source journal title, year of publication, total number of citations, correspondence address (to determine which hospital the article should be attributed to), language of the original publication, and type of publication (Figure 2). All retrieved papers were exported and saved. “.CSV” format.

Similarly, mutual support is extended with regard to the expenses associated with “R&D expenditure”, which refers to the cost of R&D. An explanation for this is provided below.

The list of publications from nine different affiliations labeled as “ISSSTE,” by Scopus was downloaded separately and then merged into a single list. Some papers were authored by two or more hospitals and thus appeared multiple times in the merged list. We manually eliminated these duplicates to avoid counting them multiple times (Figure 2).

One of the major players in Mexican research is the 13 Institutos Nacionales de Salud (CCINSHAE), which primarily focus on organ-oriented institutions (heart, lung, brain disorders, etc), to facilitate further comparisons with these institutes, we manually classified each paper according to their topic (Figure 2) using 12 subcategories (selected based on the focus of these Institutos): internal medicine and surgery (INCMNSZ), public health (INSP), obstetrics and gynecology (INPER), pediatrics (INP and HIMFG), psychiatry (INPRF), neurosciences (INNN), geriatrics (INGER), cardiology (INCICH), infectious and respiratory diseases, including ENT and critical medicine (INER), rheumatology/Trauma and Orthopedics (INR), genetics (INMEGEN) and oncology (INCAN).

To track the development of ISSSTE research over the years, we used our Excel database to organize, count, and create a graphical representation of the total publications per year from 1969 to 2021 (Figure 3).

In contrast, to illustrate the work of Mexican researchers, their approach, and their impact, we extracted the following bibliometric indicators: the number of papers, the number of citations, the mean citations per paper, and the L0 index. The L0 index, proposed by Lee (2021), is calculated as the percentage of articles not cited over the total number of published articles; in other words, it represents the percentage of non-cited papers [3]. As in any bibliometric analysis, we tracked the most cited publications and leading authors (as indicated by Scopus). We identified the most cited articles by sorting them from highest to lowest, and similarly, we sorted the most prolific authors based on their h-index and number of publications.

We also identified the journals with the highest impact factors that published papers from the institution. In our database, we located every journal that publishes Mexican literature, recorded its impact factor (IF) (Journal Citation Report consulted from Web of Science), ordered them from highest to lowest, and selected the top ten journals. It is essential to be realistic and to acknowledge that there is no shame in doing so. Identifying one’s starting point and establishing attainable objectives are commendable initiatives that can facilitate improvement. Furthermore, it is imperative to recognize that progress cannot be achieved unless it is quantified.

Regarding the nine institutions affiliated with ISSSTE, as shown in Table 1, with ISSSTE being one affiliation itself, we searched for each institution individually in the Scopus database to determine how many publications out of the total are attributed to each institution. This allowed us to understand how scientific investigations are distributed in Mexico and how authors and institutions interact. We investigated their addresses and used Excel to create geographic map charts to display the distribution of publications.

We also created an authorship network to illustrate the interactions among authors from the same institution. We used VoS Viewer (version 1.6.16) and bibliometric analysis, employing built-in utility Biblioshiny (RStudio 1.4.1106, RStudio Inc., Boston, MA, USA) for descriptive and quantitative analysis of the data. The Biblioshiny interface was utilized to identify data for concurrence and collaboration networks, co-word analysis, and word clouds (Figure 4), created with Bibliometrix 4.0.1.

To capture the content and scientific concepts presented in articles (main research subjects, we generated a tree map (using R Studio Bibliometrix package) with parameters extracted from keywords plus, author keywords, and the 50 most frequently used words in the abstracts listed in the “.CSV files” downloaded from Scopus and organized using Biblioshiny (Figure 5).

## 3. Results

By adding all papers retrieved from Scopus and with at least one author from the nine institutions indexed in the database from the ISSSTE during the period spanning 1969 (the year of the appearance of ISSSTE’s first publication) to 2021, a total of 3210 titles were downloaded and listed in our Excel file, and 1147 were manually eliminated as they were duplicated as they were written and submitted under the name of more than one affiliation and therefore appeared several times in the merged list, leaving a total of 2063 publications to analyses from ISSSTE’s whole institution profile (the same number of publications displayed during our original Scopus database research). A PRISMA diagram representing how the papers were selected is shown in Figure 2.

Prior to our publication, the interest of the scientific community on this topic was so low that in the last 70 years, we could only find five papers in PubMed with “Mexican publications” or “publications Mexico” in their title [20]: one dates back to the 1950s [21], two are from the 1970s [22,23], and the last two were written in the 1990s [24,25]. ISSSTE´s PRISMA diagram representing the paper selection process is described in Figure 2. To summarize, from 1969 (first publication) to 2021, a total of 2063 were included in the final sample from the nine institutions displayed in Table 1.

First, ISSSTE’s first article in the Scopus database was published in 1969, and for many decades, publications remained scarce with fewer than 50 publications per year. It was not until 2005 that the publication mean reached 50. As shown in Figure 3, despite a modest beginning in the sixties, it was not until 2010 that the first significant increase occurred; since then, scientific production has grown exponentially. In 2021, a peak of over 200 publications was reached, demonstrating the continued growth and expansion of research output from ISSSTE.

We identified one primary affiliation (Scopus affiliation ID:60001570) that contained articles authored by individuals affiliated with ISSSTE (our primary inclusion criteria) but not specifically attributed to a particular hospital for the ISSSTE system, respectively. Additionally, we identified eight other ISSSTE affiliations where the authors specified their hospitals.

Table 1 provides a summary of the institutions included in our analysis, along with their corresponding Scopus affiliation ID and a link to the number of documents and authors associated with each institution. We calculated the mean number of documents per author for each institution. Our analysis revealed that only four of the nine institutions had authors who collaborated on one or more documents, while authors from the other five institutions did not even reach a single collaboration.

The hospitals included in our analysis were as follows:

1. Centro Médico Nacional 20 de Noviembre; 2. Hospital Regional Lic. Adolfo López Mateos; 3. Hospital Regional 1º de Octubre; 4. Hospital Regional Dr. Valentín Gómez Farías (located in Zapopan, Jalisco); 5. Hospital General Dr. Darío Fernández Fierro; 6. General Hospital Tacuba; 7. Hospital General Dr. Fernando Quiroz and 8. Hospital ISSSTECALI (Tijuana, Baja California Norte).

It is worth noting that Scopus has misattributed affiliation 60095173 (Hospital General Dr. Fernando Quiroz) to a hospital with the same name, (Hospital General Fernando Quiroz Gutiérrez) in another state. Further information can be found at: https://www.scopus.com/affil/profile.uri?afid=60095173.

Table 2 displays the 20 most prevalent pathologies observed during medical visits to the institution, with respiratory disorders and infections, such as COVID-19 (recently ranked fifth), urinary tract infections, and gastrointestinal infections were the most prevalent. Chronic degenerative and metabolic diseases such as arterial hypertension, type 2 diabetes, and obesity are also frequently observed. In contrast, Figure 4 illustrates the 50 most frequently occurring keywords in research papers authored by the ISSSTE including terms such as “Mexico”, “treatment”, “obesity”, “cancer”, “COVID-19”, “asthma”, “mortality”, “breast cancer”, “epidemiology”, “risk factors”, “children”, “diagnosis”, “hypertension”, “diabetes” and “infection” among others. Figure 5 shows the co-occurrence of keywords (MeSH terms) using a VOSviewer in the corpus of research papers, with prominent terms such as “human”, “female”, “male”, “article”, “Mexico”, “clinical article”, “case report”, “pregnancy”, “treatment”, “child” and “risk factor”.

Figure 6 displays the subclassification of all papers according to medical specialties, providing a panorama of the areas that have been most active in terms of publications. The most prominent specialty was internal medicine and surgery (831 papers), followed by public health (499 papers), oncology (332 papers), cardiology (313 papers), neuroscience (275 papers), obstetrics and gynecology (275 papers), trauma, orthopedics, and rheumatology (249 papers), genomics (218 papers), pediatrics (203 papers), respiratory diseases and intensive care (149 papers), psychiatry (65) papers), and geriatrics (35 papers). It is important to note that each article could be included in more than one specialty depending on the topic.

Most publications analyzed in this study were originally written in Spanish (52%), followed by English (43%) or a combination of both languages (7%). French (0.09%) and German (0.94%) were also represented, albeit to a lesser extent. In terms of document type, original articles were the most common (84%) followed by reviews (8%), letters (3%), conference papers (2%), book chapters, editorials, notes, and short surveys (1% each).

Table 3 lists the top ten most cited authors affiliated with ISSSTE, ranked by their h-index. These authors were affiliated with the ISSSTE as a general institution, Hospital Regional 1º de Octubre, and Hospital Regional Adolfo López Mateos. Fiacro Jiménez-Ponce is identified as one of the authors who contributed the most to the scientific production of ISSSTE, with a total of 55 publications and the highest h-index of 25. Juan-Antonio González-Barrios from Hospital Regional 1º de Octubre had 50 publications and an h-index of 15. The two authors with the lowest h-index are from Hospital Regional Adolfo López Mateos, both with an h-index of 9, but with 28 and 14 publications, respectively.

Table 4 presents the most highly cited publications along with the institutions with the greatest impact. CMN 20 de Nov leads with the top-cited publication: “Celecoxib versus naproxen and diclofenac in osteoarthritis patients: SUCCESS-I study”, which has 268 citations). Table 4 also lists 11 other articles with more than 100 citations, with the lowest being 104 and the highest being 355. Three of these articles were published by CMN 20 de Nov, including the oldest publication on the list “Demonstration of central γ-aminobutyrate-containing nerve terminals by means of antibodies against glutamate decarboxylase”. It is worth noting that while the most cited article has its author affiliated with CMN 20 de Nov, the most cited article with ISSSTE as the mother affiliation is “Evolution of mortality over time in patients receiving mechanical ventilation” with 355 citations. As shown in Table 1 and Table 4, ISSSTE remains the most prominent mother affiliation with 1844 documents.

Figure 7 demonstrates that most of the scientific research produced by the ISSSTE is centralized in Mexico City, accounting for 92.4% of all publications. The remaining publications were published in Jalisco (6.2%) and Baja California (1.3%).

Figure 8 reveals that among the 92.4% of scientific research produced in Mexico City, the borough with the highest scientific production is Benito Juárez, the most productive author (affiliated with CMN 20 de Nov), and Hospital General Dr. Darío Fernández Fierro is also located in the same borough. Consistent with these findings, Table 1 shows that CMN 20 de Nov is the most productive institution for scientific research, with the largest number of authors (458), documents (737), and a higher document/author mean.

Figure 9 displays the top ten journals publishing Mexican papers in terms of Impact Factor (IF) The highest-ranked journal is the Journal of Clinical Oncology with an IF (of 44.54).

Table 4 presents the most cited publications in journals that correspond to the most productive journals, including “The American Journal of Respiratory and Critical Care Medicine” (n = 355), “The American Journal of Medicine” (n = 268), and “The International Journal of Clinical Practice” (n = 206). The number of citations ranged from 355 to 0, with an L0 index (proportion of uncited items relative to the total number of publications) of 59.8% (1235/2063). Of the publications, 8% were published in national journals, while the remaining were published in international journals.

Figure 10 illustrates the co-authorship network generated using VOSviewer version 1.6.16. We used articles downloaded from. “.CSV” format from selected hospitals to analyze the relationships among authors from different institutions, highlighting the most productive authors (i.e., the most cited), those who collaborated closely with authors from other institutions, and those who worked in isolation. The size of each node in the network is directly proportional to the author’s productivity, as measured by citations. VOSviewer detects collaborations between authors from different affiliations and represents those with narrower working networks using shorter link lengths. The institutions are color-coded as follows: dark green (Centro Médico Nacional 20 de Noviembre), brown (Hospital Regional 1o de Octubre), pink (Hospital Lic. Adolfo López Mateos), red (ISSSTE), yellow (Hospital General Dr. Darío Fernández), light green (Hospital General Tacuba), blue (Hospital General Dr. Fernando Quiroz) gray (ISSSTECALI Tijuana) and light pink (Hospital Regional Dr. Valentín Gómez Farías). The orange, light blue, purple, and light purple nodes represent authors who are less productive (i.e., cited less frequently). The VOSviewer program arranged the nodes to display the management correlations and summarize the research obtained.

## 4. Discussion

As mentioned in the introduction, bibliometric analysis (BA) has largely been ignored by Mexican scientists in the medical field. In other parts of the world, BA has sparked a lot of interest, and developed nations (from North America [16], Europe [17], Asia [20], and Oceania [21], as well as less advanced economies in South America [22] and Africa [23], have made efforts to evaluate the importance of their contributions to the global corpus of science. We would like to encourage other Mexican researchers to join our efforts to analyze our contribution to science.

As philosopher Arthur Schopenhauer once said, “Consciousness is awareness of awareness [24].” If we are unaware of our shortcomings, can we change them? Our goal is to raise awareness in the Latin American medical community about the importance of knowing where we stand so that we can identify the right direction for future progress.

We see our bibliometric analysis as a type of Google Map for research [25]. Like a web mapping platform, it provides an aerial view of research [26], which can help policymakers identify neglected areas (analogous to knowledge gaps) [27], pinpoint where researchers are most active (analogous to areas of scientific interest) [28], and predict where future research should focus [29], such as on the most prevalent pathologies in medical consultations.

Beyond providing healthcare services, ISSSTE also plays a role in education and scientific research. The generation of new knowledge (the goal of medical research) and its implementation are mandatory to promote wellness and offer patients a better quality of life. The core of medical progress is the development of new technologies or procedures; however, social and health policies are also necessary to implement them in different circumstances (translational research). Research must be implemented and encouraged, as it is a turning point for problem-solving and improves the training of human resources who are capable of providing better healthcare. As with many other Mexican healthcare institutions, ISSSTE intends to define and create new strategies to promote scientific research on the most prevalent and priority issues. The purpose of our study is to disclose who are the main stakeholders of Mexican medical research and how often the specialties dealing with the most prevalent pathologies contribute to the corpus of world medical knowledge. Indeed, any research project that does not culminate in a publication (by definition, accessible to the public) remains private (only known by the researchers). In our bibliometric analysis, by focusing on the most cited papers, we can determine what has drawn the attention of this medical community and compare them with the most prevalent health conditions treated by the ISSSTE.

The first question is whether health system research in Mexico focuses on patient needs. Although we cannot provide a definitive answer, we hope that our curiosity will inspire others to build on our initial attempts. However, we discovered some interesting data in our research. By comparing the keywords in Table 2 and Figure 4, such as “hypertension”, “COVID-19”, “asthma”, “obesity”, “diabetes” and “infection” we can see that authors are indeed interested in and writing about the most prevalent pathologies in their clinical practice. Nonetheless, when comparing the main disorders requiring medical care in the ISSSTE (Table 2) and the most cited articles (Table 4), it becomes evident that some highly prevalent conditions such as metabolic syndrome, dyslipidemia, hypertension, atherosclerosis, gastrointestinal conditions, pregnancy, and genitourinary infections are not given priority in the top papers, although they appear as the 20 most prevalent pathologies and the 50 most frequently occurring keywords in papers authored by the ISSSTE. However, topics related to respiratory diseases, including COVID-19, asthma, mechanical ventilation, and respiratory distress syndrome, are among the most cited articles, even though they are not the most frequent causes of medical care. Resolving this issue is beyond the scope of our study, and requires further research.

The number of publications by an author can be influenced by various factors, ranging from exceptional skills and collaboration to the unethical practices of adding undeserving authors to a paper [30]. The issue of honorary authorship—those listed and not having participated in the work needed to publish original research—is a controversial topic that has been reported in several research fields, including radiology, where the department head was sometimes automatically listed as an author (with undeserved credit in 26% of the publications) [31]. We do not know if some of these top Mexican scientists have honorary authorship [32], even if they did not contribute to the research work. This phenomenon, known to everybody, has been recently termed the “add-my-name” practice [33] and involves researchers adding their colleagues’ names as co-authors to research articles without any intellectual contribution. Sometimes, this is done in reciprocal form. That is, the authors have a network of friends who add each other’s names in turns. Further studies are needed to determine whether this practice exists in Mexico and how often it might occur.

The majority of national scientific research in Mexico is concentrated in Mexico City, with the most productive authors affiliated with CMN 20 de Noviembre. However, some research centers have very few publications, and the ratio of documents per author is low, which could suggest a lack of interest in participating in the publication process or overcrowding of authors for a single paper. This trend should be investigated further.

During our research, we discovered an issue with affiliation names that have not been previously reported. We found that two hospitals from different institutions share the same name (Fernando Quiroz). One of the hospitals was affiliated with the ISSSTE, while the other was not. The former is located in the Xico Valle de Chalco district near the suburbs of Mexico City. However, we noticed that all papers affiliated with ISSSTE’s Fernando Quiroz are misattributed to “Hospital Xico Valle” by Scopus (ID:60095173), which is listed in the corresponding neighboring suburb. This means that none of the research or teaching departments had noticed this error in disambiguation, let alone requested that Scopus correct it. Given this, we strongly recommend that heads of research or teaching departments search for their own institution’s name and request corrections from Elsevier. Scopus builds institution profiles from the affiliation text in the articles it indexes, using algorithms to identify name variations and link them to a single profile. However, because of the wide variation in how affiliations are represented by authors and publishers, this automated process is not exact. Some variants may have been missed, whereas others may have been inappropriately included. Authorized users knowledgeable about the institution can manually review alternate names to ensure that the profiles are correct and complete. Updating institutional affiliations: Scopus limits access to the Institutional Profile Wizard (IPW) to the institution’s authorized representatives. Only these users can access the profile and hierarchy of an institution.

It is important to note that various organizations that rank and evaluate institutions use Scopus data for their calculations. Institutions often use the same data for reporting, bibliometrics, and other purposes. Changes to profile information can alter an institution’s hierarchy and document counts, which can impact any institutional evaluation that uses Scopus data. Therefore, it is crucial to have authorized representatives make changes via IPW.

Our co-authorship analysis (Figure 10) reveals how research projects are shared between hospitals that belong to a single network: ISSSTE. This network diagram clearly illustrates how some institutions are closely linked, while others operate independently and “function” as independent contributors in research. The analysis confirms that the most productive authors belong to the research centers ISSSTE, and CMN 20 de Nov, as indicated by the larger nodes (red and dark green, respectively) and the most co-authored links. Interestingly, some hospitals do not appear to have tight connections, such as Adolfo López Mateos Hospital, Tacuba General Hospital, and ISSSTECALI. In contrast, the Gustavo A. Madero borough has 176 publications that are attributed to a single hospital: Hospital Regional 1º de Octubre. Although this hospital has many nodes, it has few co-author links. It appears that some institutions do not collaborate effectively with other authors despite being under the same administration.

This can create barriers to knowledge distribution. A potential reason for the limited impact of Mexican medical literature on the field of research is the presence of a constrained research and scientific inquiry culture within the medical community, which impedes the creation and dissemination of innovative ideas. Other contributing factors may include restricted access to funding, challenges presented by language barriers, and obstacles associated with publishing research in high-impact journals due to insufficient visibility and recognition in the global academic community.

Furthermore, our L0 index, which is a bibliometric parameter that reveals the significance of the Mexican medical literature, was relatively high. Upon analyzing Figure 10 and Table 1, we observed that the number of documents listed in Table 1 is more than 2063. This indicates that some publications include authors from different affiliations, promoting co-authorship and teamwork between institutions.

We used Scopus rather than the Web of Science (WOS) for our analysis. First, it has been established at the international level that Scopus is a curated, high-quality bibliometric data source for academic research in quantitative science studies [34]. Second, it has broader coverage than the Web of Science, which allows for a better representation of the sampled literature [35]. In Mexico, the National Council of Science and Technology (CONACYT) evaluates scientific journals using the Mexican Science and Technology Journal Classification System (SCRMCYT). Of the 35 Mexican journals included in this list, 89% were indexed in Scopus, 49% in PubMed, and 34% in WOS [36]. Selecting another repository would greatly underestimate Mexican production. Similar studies have compared the coverage of different databases in other countries. For example, a recent study conducted in India found that the Web of Science database indexed only 1025 conference papers, compared to 20,189 in Scopus and 21,182 in Dimensions [37]. Therefore, one of the main reasons for the variation in publication volume is the variable coverage of conference papers in the Web of Science and the other two databases.

We believe that a nation’s scientific output, both the individual and collective actions of its scientists, reflects their views on knowledge generation, how they conceive this process, and the importance their society places on research. The cross-pollination of disciplines is fundamental to truly revolutionary advances in any given field, and medicine can no longer ignore scientometrics.

To explain this, we “considered” R&D expenditure, which refers to the R&D cost of research and development. According to Forbes, it is considered one of the main engines of a country’s economy and is a fundamental component for the generation of talent and innovation [38]. Currently, global spending on R&D has reached a record high of almost USD 1.7 trillion, unfortunately, the top 10 countries account for 80% of this spending. As part of the Sustainable Development Goals (SDGs), countries have pledged to substantially increase public and private R&D spending as well as the number of researchers by 2030.

Investing in science today will make any country richer. Increasing investment in R&D will accelerate the development of these countries. In contrast, if a country skimps on resources for R&D or basic science, it will fall behind in world rankings, losing the talent of its professionals as they flee to nations that invest in them and their projects [39]. Therefore, not only publishing more in terms of quantity but also in terms of quality (by conducting projects that are relevant to our country and truly original and disruptive research, instead of replicating studies already published abroad), is critical. This will reduce reliance on foreign technology and help our nation progress towards independence from developed nations.

The positive relationship between continual investment in research and development (R&D) and economic growth has been known for many years. However, Figure 1 shows that there is a constant lack of future investment in developing countries, as demonstrated by our research. Even though the percentage of investment in R&D is higher in developed countries than in developing countries such as ours, it is still not sufficient to support future growth.

It is important to mention that some data are not available from the World Bank source; therefore, some data points are missing in Figure 1. However, Appendix A shows a higher R&D investment in some leading countries, highlighting three large geographical areas: Europe (Switzerland, Sweden, and Austria), North America, and Southeast Asia (Korea, Japan, and China). This type of analysis is crucial to understanding where we are heading, especially when resources devoted to research are scarce, as is the case in most Latin American countries, as shown in Figure 1.

Upon closer examination of R&D figures in Latin America (Table 5) [7,40,41,42,43,44,45,46,47,48,49,50,51,52], we realized that Brazil does not have as many universities as Mexico, but performs much better in terms of the number of publications. This led us to reflect on how universities and hospitals contribute to the generation of new knowledge. Currently, it is widely known that universities have three main goals: 1. to teach and train students in their future professions, 2. to inform the public about recent advancements in science, and 3. to generate new knowledge through research. While most Mexican universities fulfill the first two roles, it remains unclear if they are contributing to the creation of new medical knowledge.

This study is the first step toward answering this crucial question. Additionally, it is worth noting that many students from other Latin American countries, including undergraduate and postgraduate medical students, come to Mexico to receive training. Therefore, it is important for Mexican universities and hospitals to reflect on their contributions to the generation of new knowledge and strive to improve this area.

We would like to clarify that our study is a pioneering analysis of Mexican medical literature, the first of its kind. Although it does not represent all Mexican literature, it covers all specialties and institutions, with nationwide distribution. Its aim is to serve as a first step in the long path to answering a crucial question: What are Mexican contributions to the global corpus of medical knowledge? Our findings have set a reference point for future comparisons and encourage further research to disclose the factors that influence both productivity and impact.

The goal of our study was to offer a real perspective on selected research topics and lead to fruitful future bibliometric research that could play an essential role in the development of Latin American countries.

We would like to emphasize that the objective of bibliometric analysis is strictly to analyze. This research does not intend to place any negative labels on publishing in less prestigious journals or publishing less than expected. We believe that, in some Latin American countries, there is a feeling of discomfort or even shame when pointing out weaknesses, such as our performance in science. However, we believe that there should be no shame in being realistic. Knowing where you stand and setting achievable goals for improvement is a great step.

It is imperative to keep in mind that you cannot improve unless you understand your status, and you cannot change your goals if you do not properly set them.

Therefore, our study aims to offer a realistic perspective on Mexican medical literature, with the intention of helping set achievable goals for improvement. We hope that our findings will encourage further research and inspire a positive attitude towards improving scientific productivity and impact in Mexico and other Latin American countries.

At present, we cannot determine the “expected performance” of Mexican medical literature. Our study may serve as a milestone for future bibliometric research in Mexico and other Latin American countries. By providing a reference point for comparison, we hope to encourage further research and promote the development of scientific productivity and impacts in the region.

Within the scientific community, the term “prolific author” is often used to describe an impressive and transcendent character. Although no threshold has been defined, it typically ranges from 40 articles in the last five years to 18 papers per year, depending on the specialty. However, while in critical care not a single author from Latin America, was labeled as “prolific”, in orthodontics, 19.2% of prolific authors were from Brazil, surpassing richer countries such as Italy or much larger, countries such as India [53,54,55].

The factors that influence arts in the world [53], but this topic has not been properly and widely studied in Latin America. Furthermore, larger countries, such as Brazil and Mexico, have economic, cultural, and social contrasts that merit different studies. Further research in this area could shed light on the unique challenges and opportunities faced by Latin American authors that have promoted the development of scientific productivity and impacts in the region.

Understanding the goals, academic profiles, and even personalities of prolific authors is highly useful for the scientific community and work in the same country and institutions as many of us, and they likely face the same challenges, yet they can publish a large number of papers that many of us cannot. While we recognize that our study may touch on a raw nerve, we believe that it is important to investigate these issues to identify the key factors that contribute to academic success.

A national survey could be a notable contribution to this initiative, helping elucidate the Mexican recipe for academic success. By understanding the factors that drive successful authors, we can promote scientific productivity and impact in Mexico and other Latin American countries.

Political or administrative measures are often palliative and short-term, as they tend to change with each new decision-maker term. The only lasting solution to the skyrocketing scientific output of the Latin American medical community is a total rewiring of our mindset. We firmly believe that if inter-institutional alliances are formed to strengthen local research and promote the exchange of information, the national performance indicators will improve. Scientific collaboration between international institutions has been shown to increase the likelihood of publication in more prestigious journals, reach a larger audience, and attract more citations, ultimately resulting in a greater impact. By promoting collaboration, we can work towards improving the quality and quantity of research produced in Latin America.

As a take-home message, we can conclude that this analysis can be taken as a call to action for every doctor who has not published and, as a result, cannot be analyzed from a bibliometric point of view, remaining “non-existent” for medical research. All research that has not been published, even if it has been conducted, remains private. However, when it is published, it becomes public and contributes to the advancement of medical knowledge. We extend an invitation to all members of the medical community in Latin America to join our efforts to build a solid group of researchers and lay a strong foundation for the future of science in our country. This can be achieved through an objective analysis of the weaknesses of our past combined with our strengths from the present. Let us work together toward a brighter future for medical research in Latin America.

Our research revealed numerous issues that require urgent attention, and their solutions cannot be achieved by waving a magic wand. It is imperative that multiple stakeholders exponentially increase efforts to address these issues. In fact, our study revealed more unanswered questions that needed to be addressed than answers to our initial inquiries. It is time for concerted efforts to address these pressing concerns.

More bibliometric analysis is needed to fine-tune our comprehension of the interests of Mexican researchers and their relationship with patients’ needs.

The value of bibliometric analysis for a specific country should be explored in more detail, as we did, by examining how hospitals interact, which specialties are producing knowledge, and those that need help to publish more. Which diseases are being studied without bringing much useful knowledge and only replicating studies with questions that have already been answered elsewhere? Moreover, what are the real health problems that researchers neglect owing to a lack of interest, opportunities, or allocated funds? It is crucial that we redirect our focus toward these neglected areas in order to make significant progress in tackling the most pressing health concerns. This detailed analysis, beyond a mere catalog of “how many articles, how many citations,” will add the necessary value to this type of research for patients.

Other aspects, not studied by the present research, also need to be studied, and we have already addressed this topic in a recent paper [15]: In highly cited papers with Mexican collaboration, are we bringing ideas or just adding patients to increase the number of subjects in an international cooperative study? The level of our research and science will increase with ideas genuinely generated by us.

Many authors have contributed only a small number of papers. Why? Do we have projects involving a large amount of work or do we have a large number of authors needing projects on which they did not work at all? This is a sensitive issue, and further studies are required on this crucial topic. The prevalence of unjustified authorship has not yet been addressed.

The L0 index of Mexican research seems elevated (we have many uncited papers, even by our own authors), and the reasons for this must be investigated: Are the topics not relevant to other authors? Are our papers insufficiently visible? Should we target “better” foreign journals with a high impact factor, or can we increase the impact factor of national journals where our papers can find a natural way to be published?

We recognize that our study has limitations, as we have already acknowledged. Finally, our investigation has now set a parameter to compare later BA in Mexican literature, but we have more unanswered questions that urgently need to be addressed. Nonetheless, our study has provided valuable insights into the research process, leading to more informed decision-making. By understanding our past performance, we can adjust our current strategies and work towards a better future.

## Figures and Tables

**Figure 1 healthcare-11-01725-f001:**
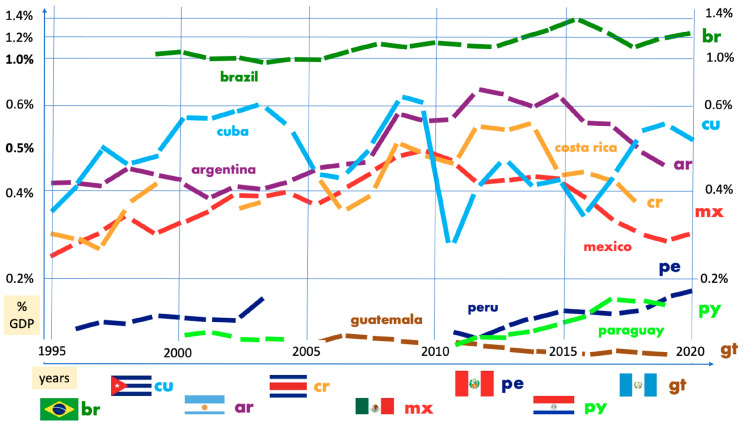
Investment in science and R&D of Latin America countries (as percentage of their GDP) from 1996–2020 with data from the World Bank.

**Figure 2 healthcare-11-01725-f002:**
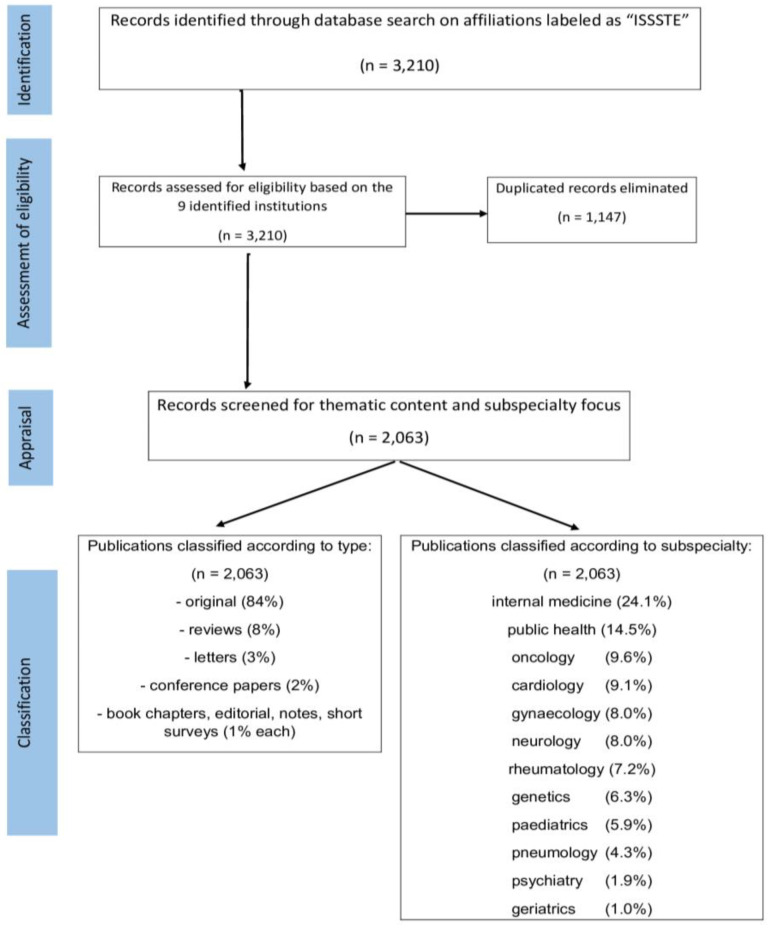
Identification of Methodology for search strategy.

**Figure 3 healthcare-11-01725-f003:**
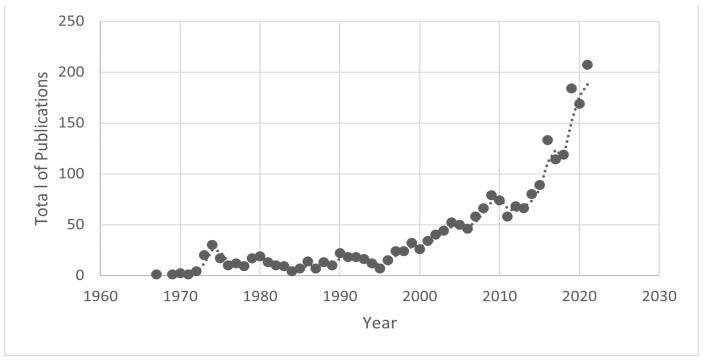
Evolution of ISSSTE’s publications with time.

**Figure 4 healthcare-11-01725-f004:**
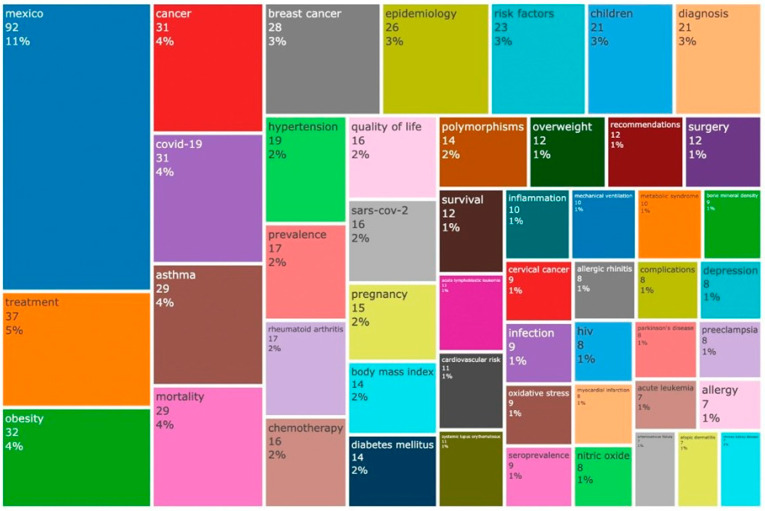
50 Most frequently occurring keywords in Mexico research authored by the ISSSTE.

**Figure 5 healthcare-11-01725-f005:**
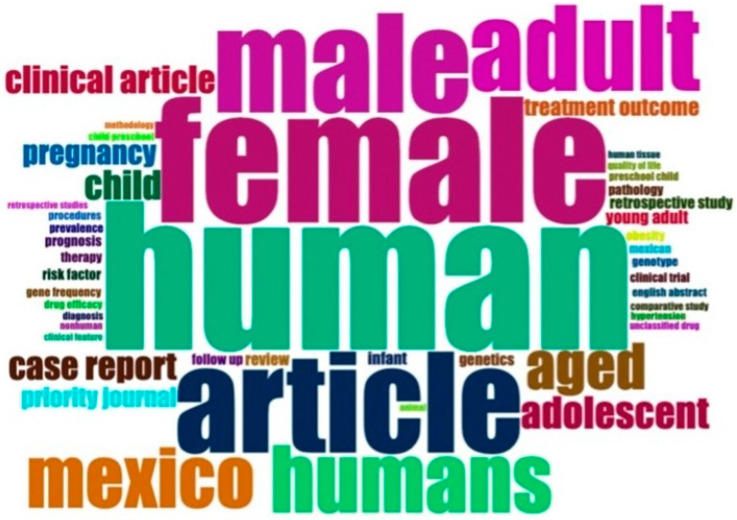
Co-occurrence of keywords (MeSH terms) using a VoS Viewer.

**Figure 6 healthcare-11-01725-f006:**
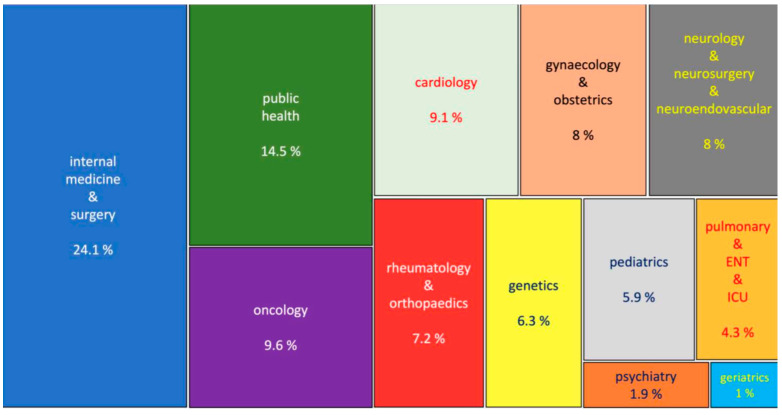
Total publications by specialty within the main institutes for health care and research.

**Figure 7 healthcare-11-01725-f007:**
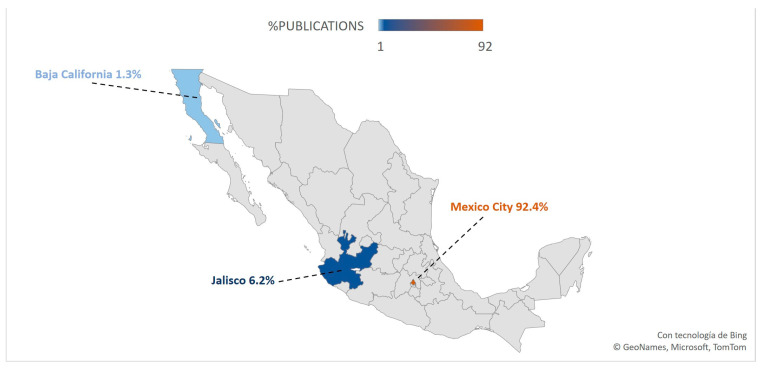
Geographic distribution of publications with at least one author affiliated to the ISSSTE.

**Figure 8 healthcare-11-01725-f008:**
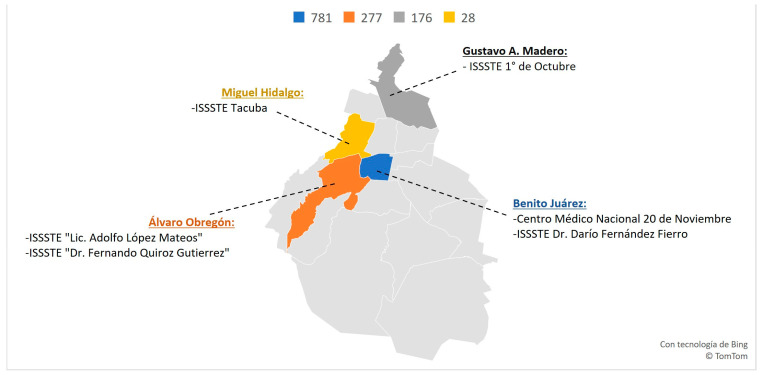
Scientific distribution among ISSSTE’s institutes in Mexico City. Distribution by City Boroughs.

**Figure 9 healthcare-11-01725-f009:**
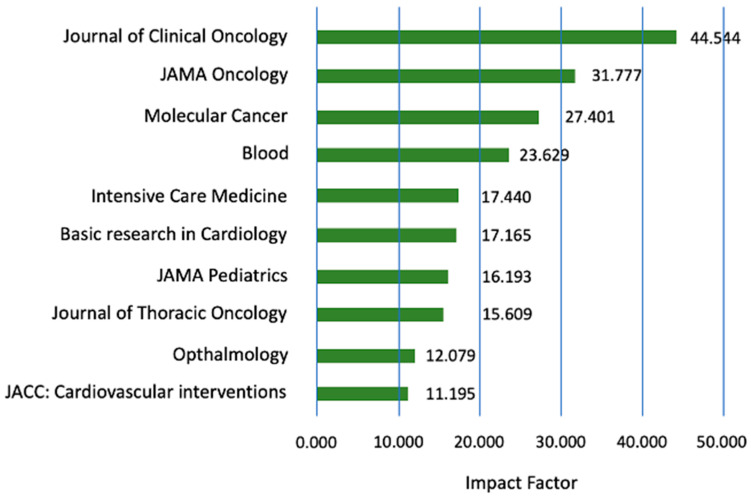
Top 10 index journals in Scopus with the highest impact factor during the period from 1969–2021.

**Figure 10 healthcare-11-01725-f010:**
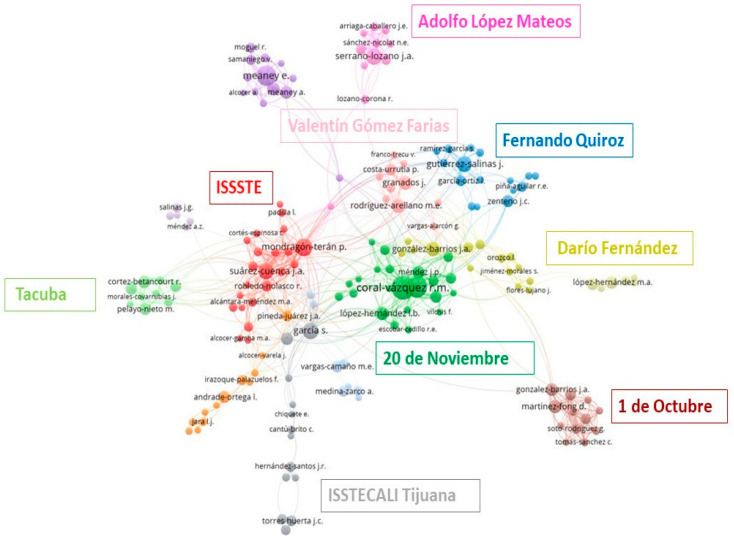
Co-authorship network between ISSSTE’s affiliated institutions.

**Table 1 healthcare-11-01725-t001:** Institutions belonging to the ISSSTE with articles indexed in Scopus.

Institution Name	Affiliation ID	Documents	Authors	Documents/Author (Mean)
ISSSTE	60001570	1844	1688	1.09
20 de Noviembre	60000123	737	458	1.6
HR “Lic. Adolfo López Mateos”	60095121	253	295	0.85
HR 1° de Octubre ISSSTE	60094760	176	117	1.5
HR Dr. Valentín Gómez Farías	60094733	85	126	0.67
HG Darío Fernández Fierro	60095178	45	48	0.93
HG Tacuba	60095176	28	25	1.12
HG Dr. Fernando Quiroz Gutierrez	60095173	24	47	0.51
Hospital ISSSTECALI, Tijuana	60095202	18	32	0.56

**Table 2 healthcare-11-01725-t002:** Twenty most prevalent pathologies in ISSSTE’s medical consultation 2020.

Notification Source—Issste		
	Medical Condition	Total
1	Acute respiratory infections	696,601
2	Urinary tract infections	154,537
3	Gastrointestinal infections	153,053
4	Arterial hypertension	63,888
5	COVID-19	63,288
6	Ulcers, gastritis, duodenitis	61,707
7	Non-insulin dependent Diabetes (Type II)	58,838
8	Conjuntivitis	33,284
9	Acute otitis media	33,284
10	Periodontal disease	31,528
11	Obesity	29,133
12	Influenza	21,077
13	Pneumonia and bronchopneumonia	17,339
14	Asthma	16,719
15	Vulvovaginitis	15,119
16	Peripheral venous insufficiency	11,661
17	Strep throat and tonsilitis	8194
18	Urogenital candidiasis	6542
19	Intestinal amoebiasis	6380
20	Scorpion sting	3342
Total		1,482,439

**Table 3 healthcare-11-01725-t003:** Leading authors affiliated to ISSSTE.

Affiliation	Author	Author ID	Publications	h-Index
ISSSTE	Jiménez-Ponce, Fiacro	7103257425	55	25
1º Octubre—ISSSTE	González-Barrios, Juan-Antonio	6602186289	50	15
1º Octubre—ISSSTE	Villagómez-Asisclo, Jesús	55814022200	22	13
ISSSTE	Etuk-Saturday, J	6701731929	44	12
ISSSTE	Andrade-Ortega, Lilia	21742168900	26	11
ISSSTE	Gutierréz-Salinas, José	6603221283	46	9
ISSSTE	Suárez-Cuenca, San-Antonio	6506202108	44	9
ISSSTE	Alcaráz-Estrada, Sofía Lizeth	56047021500	36	9
“Adolfo López Mateos”—ISSSTE	Rodríguez-Arellano, Martha-Eunice	53264796000	28	9
“Adolfo López Mateos”—ISSSTE	Coronel, Jaime-Alberto	56013587000	14	9

This list has been published without the consent of authors as it is current practice in bibliometrics, and these are publicly available data.

**Table 4 healthcare-11-01725-t004:** Most cited publications in ISSSTE.

Institution	Article Title	Year	Total Citations	Journal
ISSSTE	Evolution of mortality over time in patients receiving mechanical ventilation	2013	355	American Journal of Respiratory and Critical Care Medicine
CMN 20 Nov.	Celecoxib versus naproxen and diclofenac in osteoarthritis patients: SUCCESS-I study	2006	268	The American Journal of Medicine
ISSSTE	Effect of budesonide/formoterol maintenance and reliever therapy on asthma exacerbations	2007	206	The international Journal of Clinical Practice
CMN 20 Nov.	Review of natural products with hepatoprotective effects	2014	190	World Journal of Gastroenterology
ISSSTE	Budesonide/formoterol for maintenance and relief in uncontrolled asthma vs. high-dose salmeterol/fluticasone	2007	173	Respiratory Medicine
CMN 20 Nov.	Demonstration of central γ-aminobutyrate-containing nerve terminals by means of antibodies against glutamate decarboxylase	1981	149	Neuroscience
ISSSTE	The FTO gene is associated with adulthood obesity in the Mexican population	2017	140	Nature genetics
ISSSTE	The genomic landscape of balanced cytogenetic abnormalities associated with human congenital anomalies	2008	134	Internal Journal of Obesity
ISSSTE	Update on chemotherapeutic agents utilized for perioperative intraperitoneal chemotherapy	2005	125	Oncology. International Journal of Cancer Research and Treatment
ISSSTE	Formula and nomogram for the sphygmomanometric calculation of the mean arterial pressure	2000	112	Heart Journals
ISSSTE	Maternal and Neonatal Morbidity and Mortality among Pregnant Women with and without COVID-19 Infection: The INTERCOVID Multinational Cohort Study	2021	111	JAMA Pediatrics
ISSSTE	Updated frequency of EGFR and KRAS mutations in NonSmall-cell lung cancer in Latin America: The Latin-American consortium for the investigation of lung cancer (CLICaP)	2015	106	Journal of Thoracic Oncology
ISSSTE	Severe hypercapnia and outcome of mechanically ventilated patients with moderate or severe acute respiratory distress syndrome	2017	104	Journal of Intensive Care Medicine

Abbreviation CMN 20 Nov: Centro Médico Nacional 20 de Noviembre.

**Table 5 healthcare-11-01725-t005:** Latin American countries with their GDP per capita, R&D spending, number of Universities and journals.

Countries	Per Capita GDP (US$)	Spending on R&D (%GDP)	Universities	Number of Journals
Brazil	11,200	1.21	894	62
Mexico	8346	0.5	3082	24
Cuba	100,023	0.52	65	39
Argentina	10,729	0.46	131	45
Ecuador	5934	0.44	71	15
Costa Rica	12,508	0.37	84	52
Chile	16,502	0.34	55	68
Peru	6692	0.17	51	87
Paraguay	5400	0.14	54	60
Guatemala	5025	0.3	75	6

## Data Availability

Original CSV files from each institution along with the final list of papers in Excel can be provided by the corresponding author upon reasonable request.

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
