# Peer review of "Bibliometric Analysis: Six Decades of Scientific Production from a Nationwide Institution: Instituto de Seguridad y Servicios Sociales de los Trabajadores del Estado (ISSSTE) from Mexico"

_healthcare, 2023, doi:10.3390/healthcare11121725_

Round 1

Reviewer 1 Report

See attached pdf document. 

Author Response

Dear Reviewer thanks for your comments. We made the changes that you pointed out a few days ago. We remain attentive to any other observation. thank you

Reviewer 2 Report

Title: Bibliometric analysis: Six decades of scientific production from a single nationwide institution: ISSSTE from Mexico.

Reviewer Comments: 

Summary:

In this manuscript authors have tried to evaluate ISSSTE’s scientific research performance and identify some research gaps by bibliometric analysis of the several scientific papers. Bibliometric analysis provides insight into knowledge gaps of a particular field. Authors shortlisted papers belonging to “ISSSTE” and used Scopus database for further analysis. Authors observed that internal medicine had the greatest number of papers and found that original papers represent 82% of the papers. These results will be useful for law makers to identify the neglected areas and to focus on important areas of research. 

Strengths:

1.    These types of studies are useful for the authorities to prioritize and focus on the right scientific areas to explore further.

2.    There are very few studies conducted in this area.

Weaknesses: 

1.    Figures and table quality need to be improved.

2.    Font size and style ate different for Tables and Figures.

3.    Moderate English language changes are required. 

Author Response

(The authors gave the same response as above.)

Reviewer 3 Report

I appreciate the opportunity to critically review the article entitled "Bibliometric analysis: Six decades of scientific production from 2 a single nationwide institution: Instituto de Seguridad y Servicios Sociales de los Trabajadores del Estado (ISSSTE) from 4 Mexico". In general, the article is well written and the methodology is adequately described. The information presented could be of particular interest to the workers of this particular health institution. I am only concerned about one aspect that I consider to be major but that can be addressed by the authors. References 19-28 are not directly related to the subject of this manuscript, so I suggest omitting them. Furthermore, several of them are really old (more than 40 years).

Author Response

(The authors gave the same response as above.)

Reviewer 4 Report

Thank you very much for the opportunity to review the paper. The authors have put a lot of effort into analyzing the collected material. The work is needed, and the results are worth publishing. One can feel that the analysis of the collected material was scientific and emotional for the authors. Unfortunately, emotions in some parts of the work, especially in the Discussion, take the upper hand over the scientific aspects of the work. 

Below are detailed comments that I hope will allow the authors to improve the work according to the standards of scientific work. 

The manuscript submitted for review is quite chaotic and finding a rational purpose for scientific work is challenging. Many sentence-opinions need to be supported by scientific facts, making it more qualified for a journalistic or popular science article, but not a scientific one. 

Nevertheless, I believe that the authors, with the help of my guidance, will approach their manuscript with a "cool eye" and prepare an excellent manuscript based on the data collected and compiled. 

Introduction

This part of the paper needs to be systematized and structured, and it contains much important information that should flow one from the other. It is also essential to provide sources for some of the sentences because, in this version, they are only the author's opinion and not confirmation of facts. 

The Introduction contains the following information, in order of entry:

1. description of the health system in Mexico

2. reference to the Best Health Care ranking

3. a description of ISSSTE

4. an explanation of why scientific research in medicine serves a purpose (lines: 55-59), which without references is just an opinion - and a somewhat controversial one at that

5. a sentence about innovation, which is also an unsupported opinion

6. reference to Mexico's law in terms of scientific research

7. a comparative chart for Latin American countries

8. a description of the G20 and OECD and Mexico's role in this framework

9. description of the problem of the lack of science policy in Mexico

10. characterization of bibliometrics

11. characterization of the purpose of the study

As you can see, therefore, it is chaos. As a reader, I cannot answer the question: Where does science end and journalism begin in the Introduction? And as a reviewer, I can't find an answer to the question: what questions in the Introduction did the authors want to answer?

In my opinion, it is undoubtedly necessary to introduce subsections in the Introduction, such as Characters of bibliometric research, and to arrange the parts I mentioned above into a logical sequence of events. Usually, in the Introduction, we describe events from general to specific or from global situation to national situation. I am also not convinced that putting Figure 1. in the Introduction is necessary. Will we lose anything if we abandon this Figure? 

Based on the Introduction, I cannot say why the ISSSTE research analysis is crucial and worth analyzing. Can this be made more specific? It is essential because it is the core of the study.

Other but also essential comments in the Introduction:

1. what do the authors mean by using the phrase "national medicine" in line 106 - I feel that this is not an accurate formulation

2. instead of the word "objective" wouldn't "aim" be better?

3. what does it mean that "Mexico is a leading nation in Latin America..."? (Line 126). I'm not saying it doesn't, and I would love to know what Mexico is a leader in. These types of sentences, a dozen in the Introduction, need to be supported by references. Otherwise, they are opinions for which there is room at most in conclusions or outside this type of scientific work. 

Materials and Methods

Do the authors see any opportunity to organize and number the tasks in this chapter? Were the tasks that were listed done in that order? Were any of the tasks done simultaneously? It would be helpful to arrange it according to the order of the activities undertaken and emphasize at the beginning of the chapter that this is how it is described. 

This chapter also has "journalistic" phrases like "One of (the) major players in Mexican research..."(line 159). I would undoubtedly dispense with Figure 5, which, while interesting, deviates from charts in scientific papers. 

Results

What was written in the first sentences of this chapter (lines: 207-2015) has already been indicated - as the authors rightly write at the end of this paragraph - in Figure 2. There is, therefore, no point in repeating it. 

Line 216: "the interest.... of this topic", or what? A bibliometric analysis? It should be clarified. 

Lines 220-225: in this case, too, we repeat what was contained in the previous chapter in Figure 3. Moving Figure 2 and Figure 3 from the Methods chapter to the Results chapter may make sense. Describing what is in figures or tables differs from currently accepted rules for writing scientific papers. 

In lines 226-234, we have a description of Table 1, which appeared in the previous chapter. As a reader, I am convinced that Figures and Tables are not in the chapter where they should be. 

Line 243: How do we understand the term "the real world of medicine"?

Line 255: I am partially convinced by the scientific value of the data in Table 2.

Line 249: Reference to a chart in another chapter, although a moment earlier, there was a table described in this chapter. What is the reason for this logic of placing tables? And for what reason are references to individual tables, including Table 2, not under the table or above the table but separated by a reference to a chart from another chapter? 

Line 307: In previous tables, there was a logic applied, reading from the top of the table, from most often, the most, to the least often, the least, therefore also, in Figure 9, this should be followed - at the moment it is from least to most.

Line 309-315: What is the reason for the reference here to Table 4, which is on another page?

Discussion

Line 336: Why do the authors use scientific performance notation in quotation marks in this part of the paper? Using quotation marks causes the authors to distance themselves from the purpose of the study. I agree that the scientific performance aspect is controversial, but the consideration of the controversiality of the term should be resolved in the Introduction or in Methods by introducing the operationalization of terms. 

Line 343-343: "Consciousness is awareness of awareness..." it is tough to understand this sentence/question - what purpose does it serve besides journalistic considerations at the level of rhetorical questions?

E.g., in Line 375, references to tables and figures should be capitalized in Table 2, Figure 4

Lines 353-369 - First, this paragraph should be in the Introduction rather than the Discussion. And secondly, is what is written therein derived in any way from the ISSSTE program records? If so, references should be used. 

Line 370-374: "In our country..." this sentence is pure journalism, not a sentence in an academic paper. Besides, Discussion is not the place for such considerations. Alternatively, if we know whether someone "in our country" asked this question, we write in the Introduction that someone did/did not ask this question and indicate this as a reason for us to address it. 

Lines: 430, 459, 520: there is no room for this type of question in the Discussion. Especially the latter should be answered in Methods.

Lines 526-527: shouldn't this question appear next to the research objective under Research Questions? It certainly should not appear in the Discussion.

Line 486: "Countries..." is a beautiful and laudable phrase, but do we have scientific evidence? If not, then it is journalism. 

Table 5: I need to find the rationale for putting this table in the Discussion.

Line 509-521: I need help finding the rationale of this paragraph for the value of Discussion.

Lines 532-544: These paragraphs should be a subsection: Limitations of the Study

Lines 545-582: The presented pains of Mexican scientists are very gripping but have nothing to do with scientific work. It is possible to pick out some Conclusions from this section. Still, it should be no more than five short sentences, well, and one should avoid highfalutin phrases like "And this Copernican revolution cannot come from the Holy Spirit but will emerge from a deeper insight into us, and the first step is a description of the patterns of production of our scientists." 

General editorial note: we write reference numbers in square brackets.

Author Response

(The authors gave the same response as above.)

Round 2

Reviewer 1 Report

Please see attached review report. 

Author Response

Good morning, esteemed reviewer.

We extend our sincere gratitude for the valuable time you dedicated to reviewing our manuscript. Undoubtedly, we have observed a significant enhancement in the written content, ensuring a more enriching experience for readers. Your insightful comments on our manuscript are highly appreciated. We eagerly await any further updates or feedback you may have.

Reviewer 3 Report

I appreciate the opportunity to critically review the new version of the manuscript. The authors have made a noticeable effort, and the quality of the document has improved. I believe it is suitable for publication and will be of interest to the readership of the journal.

Author Response

Good day, esteemed reviewer,

We express our sincere gratitude for your insightful remarks, which we greatly value and acknowledge as crucial in enhancing the quality of our manuscript. Undoubtedly, we have also recognized the substantial progress achieved as a result of your invaluable feedback.

Reviewer 4 Report

Thank you very much for the work the authors have put into preparing the new version of the manuscript. 

It makes it clearer and more readable.

My last small piece of advice: please think very seriously about moving Figure 5 to the Supplement. This type of figure in a scientific paper greatly weakens its scientific appeal. 

Author Response

(The authors gave the same response as above.)
